# Bit Grooming: Statistically accurate precision-preserving quantization with compression, evaluated in the netCDF Operators (NCO, v4.4.8+)

Charles S. Zender

Departments of Earth System Science and Computer Science, University of California, Irvine, Irvine, CA 92697-3100, USA

*Correspondence to:* C. S. Zender (zender@uci.edu)

**Abstract.** Geoscientific models and measurements generate false precision (scientifically meaningless data bits) that wastes storage space. False precision can mislead (by implying noise is signal) and be scientifically pointless, especially for measurements. By contrast, lossy compression can be both economical (save space) and heuristic (clarify data limitations) without compromising the scientific integrity of data. Data quantization can thus be appropriate regardless of whether space limitations are a concern. We introduce, implement, and characterize a new lossy compression scheme suitable for IEEE floating-point data. Our new Bit Grooming algorithm alternately shaves (to zero) and sets (to one) the least significant bits of consecutive values to preserve a desired precision. This is a symmetric, two-sided variant of an algorithm sometimes called Bit Shaving which quantizes values solely by zeroing bits. Our variation eliminates the artificial low-bias produced by always zeroing bits, and makes Bit Grooming more suitable for arrays and multi-dimensional fields whose mean statistics are important.

Bit Grooming relies on standard lossless compression to achieve the actual reduction in storage space, so we tested Bit Grooming by applying the DEFLATE compression algorithm to bit-groomed and full-precision climate data stored in netCDF3, netCDF4, HDF4, and HDF5 formats. Bit Grooming reduces the storage space required by initially uncompressed and compressed climate data by 25–80% and 5–65%, respectively, for single-precision values (the most common case for climate data) quantized to retain 1–5 decimal digits of precision. The potential reduction is greater for double-precision datasets. When used aggressively (i.e., preserving only 1–2 digits), Bit Grooming produces storage reductions comparable to other quantization techniques such as linear packing. Unlike linear packing, whose guaranteed precision rapidly degrades within the relatively narrow dynamic range of values that it can compress, Bit Grooming guarantees the specified precision throughout the full floating-point range. Data quantization by Bit Grooming is irreversible (i.e., lossy) yet transparent, meaning that no extra processing is required by data users/readers. Hence Bit Grooming can easily reduce data storage volume without sacrificing scientific precision or imposing extra burdens on users.

## 1 Introduction

The increased resolution of geoscientific models and measurements (GSMMs) leads to increases in dataset size that outpace improvements in both accuracy (nearness to true values) and precision (degree of repeatability). Numerical precision that exceeds true or assumed knowledge of the underlying phenomena is called false precision and a significant fraction of GSMM

storage bits archive this false precision as essentially random (and therefore hard to compress) bits that lack scientific content. Lossy compression techniques can reduce storage requirements without sacrificing scientific content by eliminating unused range and/or false precision of stored fields. We introduce a new algorithm, Bit Grooming, that preserves a specified level of precision, is statistically unbiased, retains the full representable range of floating-point data, yet requires no additional software tools or filters to read or write.

For measurements there is never a scientific reason to retain false precision, as it amounts to storing random bits. Reasons to retain false precision during prognostic integrations of geoscientific models include numerical stability, conservation checks (e.g., mass, energy, momentum), and correct treatment of threshold and resonance phenomena. There are fewer reasons to retain false precision after than during a simulation. Most GSMMs store their data as either four or eight-byte IEEE floating point numbers. IEEE Single-Precision (SP, four-byte) and Double-Precision (DP, eight-byte) formats (*IEEE*, 2008) represent six and fifteen decimal digits of precision, respectively. Even SP often exceeds the precision to which the data are trusted. Lossy data compression can exploit the gap between the precision representable by the data type (SP or DP) and the precision associated with the values to be stored.

Data compression is well-studied (e.g., *Sayood*, 2003; *Salomon and Molta*, 2010) and before attempting lossy data compression data most researchers will check whether lossless data compression adequately serves their needs. Widely used lossless algorithms are embedded in ubiquitous (and free and patent-unencumbered) tools such as *gzip/zlib* (*Gailly and Adler*, 2000), *bzip2* (*Seward*, 2007), and *lz4* (*Collet*, 2013). These tools operate on generic byte steams. Special purpose lossless compressors designed for scientific data can exploit the four-byte or eight-byte structure of floating-point data (e.g., *Isenburg et al.*, 2005; *Burtscher and Ratanaworabhan*, 2009). Temporal and/or spatial correlations in GSMM data with large-scale patterns (e.g., climate data) can further enhance lossless compression (*Liu et al.*, 2014).

The compression ratios of lossless techniques are limited by the need to recover the exact data compressed. Lossy compression (also called quantization) relaxes this requirement and can "trade-off" precision for compression. Losses acceptable with some forms of data can only be determined subjectively, as for example the quality of photographic images. In contrast, researchers can, at least in principle, know *a priori* the scientifically defensible precision of GSMMs. False precision can mislead (by implying noise is signal) and be scientifically pointless, especially for measurements. By contrast, lossy compression can be both economical (save space) and heuristic (clarify data limitations). Data quantization can thus be appropriate regardless of whether space limitations are a concern. Thus after presenting our quantitative results, we describe techniques that make Bit Grooming simple and practical.

This manuscript is organized into four more sections. Section 2 describes the lossy and lossless compression algorithms that this manuscript will intercompare. Section 3 defines the comparison metrics and evaluates the statistical properties and compression ratios of Bit Grooming. Section 4 discusses implementation features of all lossy and lossless compression algorithms in NCO, with particular focus on Bit Grooming. Section 5 summarizes our conclusions.

## 2 Methods

A primary motivation in developing Bit Grooming is to reduce the storage of climate-related datasets. We implemented and tested Bit Grooming in the netCDF Operators, NCO (*Zender and Mangalam*, 2007; *Zender*, 2008), a freely available suite of tools for manipulating data stored in the netCDF and HDF formats (*Rew et al.*, 2006; *HDF Group*, 2015) that are widely used in the geosciences for both modeled and satellite-measured data. NCO implements or accesses four different compression algorithms, one is lossless and three are lossy. All four algorithms reduce the on-disk size of a dataset while sacrificing no (lossless) or a specified amount (lossy) of precision.

First, NCO can read and write data encoded with the (lossless) DEFLATE algorithm (*Deutsch*, 2008) accessible to both netCDF4 and HDF5 (*Rew et al.*, 2006; *HDF Group*, 2015). DEFLATE is a widely-used, freely available, and efficient compression technique that combines Lempel-Ziv compression (*Ziv and Lempel*, 1977, 1978) with Huffman coding. It identifies patterns at the bit-level and always, identifies, encodes and compresses space freed by the simple Bit Shaving (setting to zero) and Bit Setting (to one) techniques described here. DEFLATE works equally well on Bit Grooming, which is simply an alternation between Bit Shaving and Bit Setting. Some users and many data centers manually DEFLATE and re-inflate netCDF3 files with *gzip* and *gunzip* respectively, so DEFLATE is effectively available for all netCDF and HDF datasets. Hence our metrics will show the volume of uncompressed data, the same data (losslessly) deflated as the base case for compression, and the same data (lossily) quantized with Bit Grooming in tandem with DEFLATE.

### 2.1 Packing

The three lossy compression algorithms NCO employs are Packing and two precision-preserving algorithms (including Bit Grooming). Packing quantizes (usually) floating-point data into a lower precision type (fewer bytes per value) that represents a much smaller range. By convention netCDF defines a linear packing algorithm that depends on two parameters (*scale_factor* and *add_offset*) (*Rew et al.*, 2005; *Caron*, 2014a). Linear packing quantizes SP and DP data into (usually) two-byte signed integers. NetCDF uses the nomenclature `NC_FLOAT` for SP (aka float32), `NC_DOUBLE` for DP (float64), `NC_SHORT` for int16, and `NC_INT` for int32. In netCDF nomenclature, packing converts `NC_FLOAT`s and `NC_DOUBLE`s into `NC_SHORT`s). Since packing works at the byte level, the space saved is usually a factor of two (`NC_FLOAT`→`NC_SHORT`) or four (`NC_DOUBLE`→`NC_SHORT`) and cannot be specified at finer levels. Packed data can be (losslessly) deflated for additional space savings.

Packing floating-point data into integers has benefits and drawbacks. The type conversion frees-up the IEEE754 exponent bits (8 bits for SP, and 11 bits for DP) which then contribute to the dynamic range of the packed integers (16 and 32 bits for `NC_SHORT` and `NC_INT`, respectively). However, integers have a much-reduced dynamic range relative to floating-point numbers. The dynamic ranges of SP and DP numbers are $\sim 10^{37}$ and $\sim 10^{308}$, respectively, whereas data packed linearly into two-byte and four-byte integers have dynamic ranges of $\sim 10^5$ and $\sim 10^{10}$, respectively. Variables packed as `NC_SHORT`, for example, can represent only about 64000 discrete values in the range $-32768 \times \text{scale\_factor} + \text{add\_offset}$ to $32767 \times \text{scale\_factor} + \text{add\_offset}$. The optimal *add_offset* parameter for linear packing is the midpoint of the data to be packed,

and the optimal *scale_factor* is the data dynamic range (i.e., maximum minus minimum) divided by $2^{16} - 1 = 65,535$ (*Zender*, 2016a). Unpacked values must cluster within a dynamic range of $\sim 10^5$ that may itself reside anywhere within the full ($\sim 10^{37}$) floating point range. Thus archived fields that meaningfully span more than five orders of magnitude (aka five decades) are not well-suited for linear packing into two-byte integers. The presence of such fields depends on the GSMM. Candidates in climate

models include aerosol number concentrations, pressure, solar heating rates, and (some) tracer mixing ratios. Astrophysical and stellar models span larger scales and are replete with such fields, e.g., plasma density, pressure, and thermal radiation.

    Another limitation of linear packing is that the precision of packed data cannot be specified or guaranteed in advance because it depends on the distribution of values to be packed. While the numeric resolution (i.e., the smallest resolvable difference) of unpacked data always equals *scale_factor*, the number of significant digits of precision depends on the dynamic range (maxi-

mum minus minimum) of values to quantize, and rapidly degrades beyond the first decade of unpacked values. To illustrate this, consider a pressure field $p$ [Pa] uniformly spanning values $0.0 \leq p \leq 65535.0$. Linear packing exactly represents integer values in this range, and quantizes all fractional values to integers. For example, $p = 1.23456$ Pa and $p = 65534.23456$ Pa would be quantized as 1 and 65534, respectively, which have one and four significant digits (*nsd* = 1 and 4 in the terminology defined in Section 2.2 below), respectively. Packing this distribution of values achieves its highest precision (four decimal digits for

two-byte integers) only for the greatest (in absolute value) unpacked value. Unpacked values of lesser magnitude lose precision at a rate of approximately one significant digit per decade from the maximum. Since the precision of linear packing degrades by about one digit per decade, only values within one decade of the maximum regularly achieve the highest possible precision (four decimal digits for two-byte integers). This is the maximum precision that packing guarantees for an arbitrary distribution of values.

Consider the same dynamic range used previously except now offset by $10^5$ (i.e., *add_offset* = $10^5$), so $100000.0 \leq p \leq 165535.0$. The previously examined values, offset by $10^5$, are $p = 100001.23456$ Pa and $p = 165534.23456$ Pa. These would be quantized as 100001 and 165534, respectively, which both have six significant digits. Thus the *add_offset* parameter can provide additional precision to unpacked values, bringing the total precision up to six digits, for some but not all distributions of values. Except where otherwise indicated in this work we state the best precision that a compression algorithm guarantees

for any distribution of values, not the best precision it can achieve for special distributions of values.

## 2.2   Precision-Preserving Compression

The other two lossy compression algorithms considered both perform Precision-Preserving Compression (PPC). The operational definition of "significant digit" in our precision preserving algorithms is that the exact value, before rounding or quantization, is within one-half the value of the decimal place occupied by the Least Significant Digit (LSD) of the rounded value.

For example, the value $\pi = 3.14$ correctly represents the exact mathematical constant $\pi$ to three significant digits because the LSD of the rounded value (i.e., 4) is in the one-hundredths digit place, and the difference between the exact value and the rounded value is less than one-half of one one-hundredth, i.e., $3.14159265358979323844 - 3.14 = 0.00159 < 0.005$.

    One PPC algorithm preserves the specified total Number of Significant Digits (NSD) of the value. For example there is only one significant digit in the weight of most "eight-hundred pound gorillas" that you will encounter, i.e., so *nsd* = 1. NSD is

the most straightforward measure of precision, and is the default PPC algorithm in NCO. Bit Grooming combines two NSD algorithms (described below) to yield more accurate statistical properties.

The other PPC algorithm preserves the number of Decimal Significant Digits (DSD), i.e., the number of significant digits following (positive, by convention) or preceding (negative) the decimal point. For example, $0.008$ and $800$ have, respectively, three and negative two decimal digits following the decimal point, and correspond to $dsd = 3$ and $dsd = -2$.

Their fundamental difference is that NSD is independent of dimensional units and DSD is not. The NSD for a given GSMM value depends on intrinsic accuracy and error characteristics of the model or measurements. The appropriate DSD for a given value depends on these intrinsic characteristics and, in addition, the dimensional units with which values are stored. Our eight-hundred pound gorilla has $nsd = 1$ regardless of whether the value is stored in pounds or in some other unit. DSD corresponding to this weight is $dsd = -2$ if the value is stored in pounds ($8 \times 10^2$ lb), $dsd = 4$ if stored in megapounds ($8 \times 10^{-4}$ Mlb).

## 2.3 Algorithms

The time-penalty for compressing and uncompressing data varies according to the algorithm. (*Silver and Zender*, 2016) show that lossless compression dominates the total compression time, and that quantization via Bit Grooming or linear Packing can actually shorten total compression time because they reduce the amount of data to compress. At least in our implementations and for the purposes of this discussion, a Number of Significant Digit (NSD) algorithm quantizes by bitmasking, and employs no floating-point math. By contrast, a Decimal Significant Digit (DSD) algorithm quantizes by rounding, and thus does require floating-point math. Hence NSD is likely faster than DSD, though the difference has not been measured.

NSD algorithms create a bitmask to alter the significand (aka mantissa or fraction) of IEEE 754 floating-point data. For instance, the bitmask for the NSD technique called Bit Shaving is one for all bits to be retained and zero for ignored bits (*Caron*, 2014b). The logical AND of this mask with the exact IEEE value produces the quantized IEEE value. The bitmask for the NSD technique we call Bit Setting is zero for retained bits and one for discarded bits. The logical OR of this mask with the exact IEEE value produces the quantized IEEE value. These algorithms assume that the number of binary digits (i.e., bits) necessary to represent a single base-10 digit is $\ln(10)/\ln(2) = 3.32$. The exact numbers of explicit mantissa bits *Nbit* retained for single and double precision values are $\text{ceil}(3.32 \times nsd) + 1$ and $\text{ceil}(3.32 \times nsd) + 2$, respectively. (The IEEE format includes a single mantissa bit that is implicit and that is not included in these counts because it consumes no memory). This is more than predicted by the simple rule that the required number of bits is $nsd \times \ln(10)/\ln(2)$. The extra bits are the (experimentally determined) overhead required to guarantee that terminal significant digits are accurate within half the minimal value of their decimal position. Once the number of bits required exceeds the IEEE SP and DP storage standards of 23 and 53 explicit mantissa bits, respectively, bitmasking is completely ineffective. This occurs at $nsd = 6.3$ and $15.4$, respectively. To guarantee preserving 1–7 digits of precision, Bit Grooming must retain $5, 8, 11, 15, 18, 21$, and $25$ explicit mantissa bits, respectively. Thus Bit Grooming (and IEEE) require DP format to guarantee $nsd \geq 7$.

The DSD algorithm, by contrast, uses rounding to remove undesired precision. The rounding zeroes the greatest number of (Base 2) significand bits consistent with the desired (Base 10) decimal precision. Our NCO implementation rounds with the internal math library `rint()` family of functions that were standardized in C99. The exact algorithm NCO employs is

**Table 1. Exact and Lossy IEEE Single-Precision Floating Point Pi**

IEEE-754 Single Precision binary representations of $\pi$ stored exactly, with three significant digits, and with three quantization algorithms.

| Sign[a] | Exponent[b] | Significand[c] | Decimal | Notes |
|---|---|---|---|---|
| 0 | 10000000 | 10010010000111111011011 | 3.14159265 | Exact $\pi$ |
| 0 | 10000000 | 10010001111010111000011 | 3.14000000 | Three significant digits |
| 0 | 10000000 | 10010010000000000000000 | 3.14062500 | DSD = 2 (Decimal Rounding) |
| 0 | 10000000 | 10010010000000000000000 | 3.14062500 | NSD = 3 (Bit Shaving)[d] |
| 0 | 10000000 | 10010010000111111111111 | 3.14160132 | NSD = 3 (Bit Setting) |

[a]Bit 0 is $s$ which IEEE-754 format uses to encode signedness as $-1^s$.
[b]Bits 1–8 form base-2 exponent $q$ in the factor $2^{q-127}$ which in IEEE-754 multiplies the significand.
[c]Bits 9–31 are base-2 significand (or mantissa or fraction) $c$ in the IEEE-754 representation of the full value $-1^s \times (1+c) \times 2^{q-127}$.
[d]Bit Grooming and Bit Shaving are identical for a single value.

$val = \mathtt{rint}(scale \times val)/scale$ where $scale$ is the nearest power of 2 that exceeds $10^{prc}$ and the inverse of $scale$ is used when $prc < 0$. For $ppc = 3$ or $ppc = -2$, for example, we have $scale = 1024$ and $scale = 1/128$. Because our DSD algorithm rounds a Base 10 integer to achieve a Base 10 precision, we call it the Decimal Rounding algorithm. The Decimal Rounding algorithm implemented in the *nc3tonc4* software tool by J. Whitaker is distinct-from but consistent-with and equivalent-to (though not bit-for-bit with) NCO's.

Maintaining non-biased statistical properties during lossy compression requires special attention. Decimal Rounding uses `rint()` to round toward the nearest even integer. Thus our DSD algorithm has no systematic bias. However, NSD algorithms use a bitmask technique that is susceptible to statistical bias. Zeroing all non-significant bits is guaranteed to produce numbers quantized to the specified tolerance, i.e., half of the decimal value of the position occupied by the LSD. However, always zeroing the non-significant bits results in quantized numbers that never exceed the exact number. Thus Bit Shaving produces a negative bias in statistical quantities (e.g., the average) subsequently derived from the quantized numbers. Likewise Bit Setting produces a positive statistical bias. To avoid bias, Bit Grooming (our new NSD algorithm) rounds non-significant bits down (to zero) or up (to one) in an alternating fashion when processing array data. In general, the first element is rounded down, the second up, the third down, etc. Hence Bit Grooming can nearly eliminate the mean quantization bias. Our Bit Grooming implementation has one exception to the rule of alternately setting and shaving bits: never quantize upwards the floating-point value of zero. This exception prevents creation of quantization fluctuations in arrays of zeros. Finally, for simplicity, our implementation of Bit Grooming always rounds scalars downwards.

To demonstrate the change in IEEE representation caused by quantization, consider again the case of $\pi$, represented as an `NC_FLOAT`. The IEEE 754 single precision representations of the exact value (3.141592...), the value with only three significant digits treated as exact (3.140000...), and the value as stored (3.140625) after NSD ($prc = 3$) and DSD ($prc = 2$) quantization (Table 1). The string of sixteen trailing zero-bits in the rounded values facilitates both byte-stream and bitwise compression. NSD and DSD algorithms do not always produce results that are bit-for-bit identical, although they do in this

**Table 2. Bit Grooming Pi**

Same as Table 1 but after varying degrees of Bit Grooming

| Sign | Exponent | Fraction (Significand) | Decimal | Notes |
|------|----------|------------------------|---------|-------|
| 0 | 10000000 | 10010010000111111011011 | 3.14159265 | Exact |
| 0 | 10000000 | 10010010000111111011011 | 3.14159265 | NSD = 8 |
| 0 | 10000000 | 10010010000111111011010 | 3.14159262 | NSD = 7 |
| 0 | 10000000 | 10010010000111111011000 | 3.14159203 | NSD = 6 |
| 0 | 10000000 | 10010010000111111000000 | 3.14158630 | NSD = 5 |
| 0 | 10000000 | 10010010000111100000000 | 3.14154053 | NSD = 4 |
| 0 | 10000000 | 10010010000000000000000 | 3.14062500 | NSD = 3 |
| 0 | 10000000 | 10010010000000000000000 | 3.14062500 | NSD = 2 |
| 0 | 10000000 | 10010000000000000000000 | 3.12500000 | NSD = 1 |

particular case when the NSD algorithm is Bit Grooming or Bit Shaving (which are identical algorithms for a single scalar value). When the NSD algorithm is Bit Setting we obtain the fifth row where insignificant bits set to one not zero.

Reducing the preserved precision of NSD-rounding produces increasingly long strings of identical-bits amenable to compression (Table 2). The consumption of about 3 bits per digit of base-10 precision is evident, as is the coincidence of a quantized value that greatly exceeds the mandated precision for NSD = 2. Although the NSD algorithm generally masks some bits for all $nsd <= 7$ (for NC_FLOAT), compression algorithms like DEFLATE may need byte-size-or-greater (i.e., at least eight-bit) bit patterns before their algorithms can take advantage of of encoding such patterns for compression. Do not expect significantly enhanced compression from $nsd > 5$ (for NC_FLOAT) or $nsd > 14$ (for NC_DOUBLE). Clearly values stored as NC_DOUBLE (i.e., eight-bytes) are susceptible to much greater compression than NC_FLOAT for a given precision because their significands explicitly contain 53 bits rather than 23 bits.

## 3 Results

### 3.1 Metrics

How can one be sure lossy data are sufficiently precise? We define several metrics to quantify quantization error. The mean error $\bar{\epsilon}$ and mean absolute error $\bar{\epsilon}^+$ incurred in quantizing a variable from true values $x_i$ to quantized values $q_i$ are, respectively,

$$\bar{\epsilon} = \frac{\sum_{i=1}^{i=N} \mu_i m_i w_i (x_i - q_i)}{\sum_{i=1}^{i=N} \mu_i m_i w_i} \quad \text{and} \quad \bar{\epsilon}^+ = \frac{\sum_{i=1}^{i=N} \mu_i m_i w_i |x_i - q_i|}{\sum_{i=1}^{i=N} \mu_i m_i w_i}$$

where $\mu_i$ is 1 unless $x_i$ is a missing value, $m_i$ is 1 unless $x_i$ is masked, and $w_i$ is the weight. The maximum and minimum errors $\epsilon_{\max}$ and $\epsilon_{\min}$ are both signed

$$\epsilon_{\max} = \max(x_i - q_i) \qquad \text{and} \qquad \epsilon_{\min} = \min(x_i - q_i)$$

while the maximum and minimum absolute errors $\epsilon_{\max}^+$ and $\epsilon_{\min}^+$ are positive-definite.

$$\epsilon_{\max}^+ = \max|x_i - q_i| = \max(|\epsilon_{\max}|, |\epsilon_{\min}|)$$

$$\epsilon_{\min}^+ = \min|x_i - q_i| = \min(|\epsilon_{\max}|, |\epsilon_{\min}|)$$

Typically $\epsilon_{\min}^+ = 0$ for quantization, since many exact values need no quantization.

The three most important error metrics for quantization are $\epsilon_{\max}^+$, $\bar{\epsilon}^+$, and $\bar{\epsilon}$. The upper bound (worst case) quantization performance is $\epsilon_{\max}^+$. The mean accuracy $\bar{\epsilon}$ indicates whether statistical properties of quantized numbers will accurately reflect the true values. However, $\bar{\epsilon}$ allows positive and negative offsets to compensate each other and conceal poor performance. $\bar{\epsilon}^+$ measures the absolute mean accuracy of quantization, so that all errors accumulate and (unlike $\bar{\epsilon}$) do not compensate. The difference between $\epsilon_{\max}^+$ and $\bar{\epsilon}^+$ indicates how much of an outlier the worst case error is.

## 3.2 Bit Grooming vs. Bit Shaving

Traditional Bit Shaving bit-shifts zeros into the least significant bits (LSBs) of true values (*Caron*, 2014b). Thus Bit-Shaving nearly always underestimates true values, and this produces $\epsilon_{\max} = 0$. Conversely, bit-shifting ones into the LSBs, a procedure that might be called Bit Setting, would nearly always overestimate true values and result in $\epsilon_{\min} = 0$. The intrinsic compression efficiencies of Bit Shaving and Bit Setting are identical. The key innovation in Bit Grooming is to alternately bit-shift zeroes and ones into the consecutive true values in an array. By alternating high with low quantization errors, Bit Grooming balances the mean quantization error. As a result, statistical operations produce less-biased results when operating on values quantized by Bit Grooming than by Bit Shaving or Bit Setting. Balanced algorithms like Bit Grooming should yield $\epsilon_{\max} \approx -\epsilon_{\min}$, $\epsilon_{\max}^+ \approx \epsilon_{\min}^+$, and $\bar{\epsilon} \approx 0$.

All three metrics are expressed in terms of the fraction of the ten's place occupied by the LSD. If the LSD is the hundreds digit or the thousandths digit, then the metrics are fractions of 100, or of 1/1000, respectively. PPC algorithms should produce maximum absolute errors less than 0.5 in these units. If the LSD is the hundreds digit, then quantized versions of true values will be within fifty of the true value. It is much easier to satisfy this tolerance for a true value of 100 (only 50% accuracy required) than for 999 (5% accuracy required). Thus the minimum accuracy guaranteed for *nsd* = 1 ranges from 5–50%. For this reason, the best and worst cast performance usually occurs for true values whose LSD value is close to one and nine, respectively. Of course most users prefer *prc* > 1 because accuracies increase exponentially with *prc*. Continuing the previous example to *prc* = 2, quantized versions of true values from 1000–9999 will also be within 50 of the true value, i.e., have accuracies from 0.5–5%. In other words, only two significant digits are necessary to guarantee better than 5% accuracy in quantization. We recommend that dataset producers and users consider quantizing datasets with *nsd* = 3. This guarantees accuracy of 0.05–

**Table 3. Error Metrics for Bit Grooming vs. Bit Shaving**

| | | | Artificial Data[a] | | | | Observed Data[b] | | | |
|---|---|---|---|---|---|---|---|---|---|---|
| | BG and BS[c] | | BGSP | BSSP | BGDP | BSDP | BGSP | BSSP | BGDP | BSDP |
| $NSD^d$ | $\epsilon_{\max}^+$ | $\bar{\epsilon}^+$ | $\bar{\epsilon}^e$ | $\bar{\epsilon}$ | $\bar{\epsilon}$ | $\bar{\epsilon}$ | $\bar{\epsilon}$ | $\bar{\epsilon}$ | $\bar{\epsilon}$ | $\bar{\epsilon}$ |
| 1 | 0.31 | 0.11 | 4.1e−4 | −0.11 | 4.0e−4 | −0.11 | 2.4e−3 | −0.11 | 2.4e−3 | −0.11 |
| 2 | 0.39 | 0.14 | 6.8e−5 | −0.14 | 5.5e−5 | −0.14 | 3.8e−4 | −0.14 | 3.9e−4 | −0.14 |
| 3 | 0.49 | 0.17 | 1.0e−6 | −0.17 | −5.5e−7 | −0.17 | −9.6e−5 | −0.17 | −5.3e−5 | −0.18 |
| 4 | 0.30 | 0.11 | 3.2e−7 | −0.11 | −6.1e−6 | −0.11 | 2.3e−3 | −0.11 | 2.7e−3 | −0.11 |
| 5 | 0.37 | 0.13 | 3.1e−7 | −0.13 | −5.6e−6 | −0.13 | 2.2e−3 | −0.13 | 6.5e−3 | −0.13 |
| 6 | 0.36 | 0.12 | −4.4e−7 | −0.12 | −4.1e−7 | −0.17 | 1.7e−2 | −0.11 | 6.1e−2 | −0.11 |
| 7 | 0.00 | 0.00 | 0.0 | 0.00 | 1.5e−7 | −0.10 | 0.0 | 0.00 | 0.1 | 0.00 |

[a]Artificial Data is $N = 1000000$ values spanning $[1.0, 2.0)$ in equal-increment steps of $1 \times 10^{-6}$.

[b]$N = 13934592$ values of the temperature field from the NASA MERRA analysis of 20130601.

[c]BG is Bit Grooming, BS is Bit Shaving, SP is Single-Precision, and DP is Double-Precision. Values for $\epsilon_{\max}^+$ and $\bar{\epsilon}^+$ are shown only once. They are identical to two significant figures for BG and BS in both SP and DP, for both Artificial and Observed Data.

[d]NSD is Number of Significant Digits.

[e]$\bar{\epsilon}$ is shown in floating-point notation for values smaller than 0.1, i.e., 4.1e−4 means $4.1 \times 10^{-4}$.

0.5% for individual values. Statistics computed from ensembles of quantized values will, assuming the mean error $\bar{\epsilon}$ is small, have much better accuracy than 0.5%. This accuracy is the most that many applications can justify.

To demonstrate these principles we conduct error analyses on an artificial, reproducible dataset, and on an actual dataset[1] of values from a re-analysis of observed weather data. Table 3 summarizes quantization accuracy for each *NSD* based on the three metrics: the maximum absolute error $\epsilon_{\max}^+$, the mean absolute error $\bar{\epsilon}^+$, and the mean error $\bar{\epsilon}$. PPC quantization performs as expected. First, absolute maximum errors $\epsilon_{\max}^+ < 0.5$ for all *prc*. We increased the exact number of bits shaved or groomed until the worst performance ($\epsilon_{\max}^+ = 0.49$ for *prc* = 3) was better than $\epsilon_{\max}^+ = 0.5$. This guarantees that Bit Grooming always produces precision that meets or exceeds the requested number of significant digits.

For $1 \leq prc \leq 6$, quantization results in comparable maximum absolute and mean absolute errors $\epsilon_{\max}^+$ and $\bar{\epsilon}^+$, respectively (Table 3). Mean errors $\bar{\epsilon}$ are orders of magnitude smaller because quantization produces over- and under-estimated values in balance. When *prc* = 7, quantization of single-precision values is ineffective, because all available bits are used to represent the maximum precision of seven digits. The maximum and mean absolute errors $\epsilon_{\max}^+$ and $\bar{\epsilon}^+$ are nearly identical across algorithms, precisions, and dataset types. This is consistent with both the artificial data and empirical data being random, and thus exercising equally strengths and weaknesses of the algorithms over the course of millions of input values. We generated

---

[1]The artificial dataset employed is one million evenly spaced values from 1.0–2.0. The analysis data are $N = 13934592$ values of the temperature field from the NASA MERRA analysis of 20130601.

artificial arrays with many different starting values and interval spacing and all gave qualitatively similar results. The results presented are the worst obtained.

The artificial data has much smaller mean error $\bar{\epsilon}$ than the observational analysis. The reason why is unclear. It may be because the temperature field is concentrated in particular ranges of values (and associated quantization errors) prevalent on Earth, e.g., $200 < T < 320$. It is worth noting that the mean error $\bar{\epsilon} < 0.01$ for $1 <= prc < 6$, and that $\bar{\epsilon}$ is typically at least two or more orders of magnitude less than $\epsilon_{max}^{+}$. Thus quantized values with precisions as low as $prc = 1$ still yield highly significant statistics by contemporary scientific standards.

### 3.3 Compressing Real Climate Datasets

PPC quantization enhances compression of typical climate datasets. The degree of enhancement depends, of course, on the required precision. Model results are often computed as `NC_DOUBLE` then archived as `NC_FLOAT` to save space, while, in our experience, observations are usually stored as `NC_FLOAT` because most sensors lack the precision required to justify `NC_DOUBLE`. We evaluated compression performance of lossless and lossy compression techniques on four datasets representative of model-simulations and satellite-retrievals. Only floating-point data were compressed. No attempt was made to compress integer-type variables as they occupy an insignificant fraction of most climate datasets.

The first dataset tested (Table 4) comes from a global aerosol simulation (*Zender et al.*, 2003) of horizontal resolution latitude $\times$ longitude $= 64 \times 128$ (i.e., 8192 gridpoints). This dataset is the smallest (35 MB, Row A) relative to the others tested, and was produced uncompressed, as is still the norm for most climate models. Weak and strong compression (BZ1 and BZ9) with *bzip2* (*Seward*, 2007) both achieve compression ratios CR $\sim 84\%$ (Rows B–C). Conversion from netCDF3 (N3) to netCDF4 (N7) imposes a small penalty on size due to the extra internal metadata used by the underlying HDF5 format (*Rew et al.*, 2006; *HDF Group*, 2015) (Row D). Both the weak and strong HDF-implementation of DEFLATE (*Deutsch*, 2008) shrink the data to CR $\sim 81\%$ (Rows E–F), slightly better than *bzip2* (Rows B–C). There continues to be little difference between weak and strong lossless compression of a given mode (*bzip2* or DEFLATE) so for brevity in the following we focus on performance with weak (DF1) DEFLATE compression (e.g., Rows H–O).

Packing SP floating point data into two-byte integers yields CR $\sim 51\%$ (Row G). Lossless compression more than halves that CR to $\sim 23\%$ (Row H). For this dataset, Bit Grooming ranges between $81 \geq$ CR $\geq 29\%$ for $7 \geq NSD \geq 1$ (Rows I-O). Table 4 shows Packing as having $1 \lesssim NSD \lesssim 4$. Packing (into two-byte integers) uses 16-bit integers, the same as the number of mantissa bits Bit Grooming uses (as discussed in Section 2.3, and including the implicit IEEE bit) to guarantee NSD $= 4$. Section 2.1 describes why linear Packing guarantees $NSD \gtrsim 4$ precision only for the greatest decade of unpacked values, and degrades to $NSD \gtrsim 1$ for the smallest decade of unpacked values.

The second dataset tested (Table 5) contains a an atmospheric GCM simulation (*Dennis et al.*, 2012) on a higher horizontal resolution unstructured grid (with 48602 gridpoints), and occupies 840 MB uncompressed (Row A). It is about 15% more susceptible to both *bzip2* (Rows B–C) and DEFLATE (Row E–F) compression than the dataset in Table 4. The reasons for this are unclear, though at $\sim 25$-times the size of the first dataset, it seems possible that the internal metadata stored by DEFLATE is more efficient with larger datasets. Packing is nearly as efficient as before (Row G), since the CR of packing is independent

**Table 4. Compression Ratios for Low-Resolution Initially Uncompressed Model netCDF3 Data**

| Row[a] | Fmt[b] | LLC[c] | Qnt[d] | Rng[e] | NSD[f] | Size[g] | CR[h] | Method[i] |
|---|---|---|---|---|---|---|---|---|
| A | N3 | [j] | - | $10^{37}$ | $\sim$7 | 34.7 | 100.0 | Uncompressed |
| B | N3 | BZ1 | - | $10^{37}$ | $\sim$7 | 28.9 | 83.2 | Bzip2 |
| C | N3 | BZ9 | - | $10^{37}$ | $\sim$7 | 29.3 | 84.4 | Bzip2 |
| D | N7 | - | - | $10^{37}$ | $\sim$7 | 35.0 | 101.0 | Uncompressed |
| E | N7 | DF1 | - | $10^{37}$ | $\sim$7 | 28.2 | 81.3 | DEFLATE |
| F | N7 | DF9 | - | $10^{37}$ | $\sim$7 | 28.0 | 80.8 | DEFLATE |
| G | N7 | - | LP | $10^{5}$ | $\sim$1–4 | 17.6 | 50.9 | Linear Packing |
| H | N7 | DF1 | LP | $10^{5}$ | $\sim$1–4 | 7.9 | 22.8 | Linear Packing |
| I | N7 | DF1 | BG | $10^{37}$ | $\sim$7 | 28.2 | 81.3 | Bit Grooming |
| J | N7 | DF1 | BG | $10^{37}$ | 6 | 27.9 | 80.6 | Bit Grooming |
| K | N7 | DF1 | BG | $10^{37}$ | 5 | 25.9 | 74.6 | Bit Grooming |
| L | N7 | DF1 | BG | $10^{37}$ | 4 | 22.3 | 64.3 | Bit Grooming |
| M | N7 | DF1 | BG | $10^{37}$ | 3 | 18.9 | 54.6 | Bit Grooming |
| N | N7 | DF1 | BG | $10^{37}$ | 2 | 14.5 | 43.2 | Bit Grooming |
| O | N7 | DF1 | BG | $10^{37}$ | 1 | 10.0 | 29.0 | Bit Grooming |

[a]Row, also labels the compression configuration in that row.

[b]Format on-disk: N3 for netCDF CLASSIC, N4 for NETCDF4, N7 for NETCDF4_CLASSIC (which comprises netCDF3 data types and structures with netCDF4 storage features like compression), H4 for HDF4, and H5 for HDF5. N4/7 means results apply to both N4 and N7 filetypes.

[c]Lossless compression method (if any) employed. Numbers prefixed by DF refer to the strength of the DEFLATE algorithm employed internally by netCDF4/HDF5, while numbers prefixed by BZ refer to the block size employed by the Burrows-Wheeler algorithm in bzip2.

[d]Quantization (lossy compression) method (if any) employed: BG for Bit Grooming and LP for default ncpdq linear packing algorithm (convert floating-point types to NC_SHORT).

[e]Dynamic range of values compressible to indicated precision.

[f]Number of significant digits retained. Similarity symbol indicates value is approximate, not guaranteed. Full IEEE single-precision has $nsd \sim 7$ and guarantees $nsd \geq 6$. Bit Grooming guarantees specified number of digits. Linear-packing achieves $nsd \gtrsim 4$ in the largest decade of unpacked values, decreasing by one digit per decade to $nsd \gtrsim 1$ in the smallest decade of unpacked values.

[g]Resulting filesize in MB.

[h]Compression ratio in %, i.e., filesize after compression divided by its original size, times one-hundred. Compression ratios reported are relative to the size of the original file as distributed (e.g., by NASA). The original files in Tables 4 and 5 were not yet compressed, and those in Tables 6 and 7 were already compressed.

[i]Compression method, if any. Supplement provides full commands to reproduce results.

[j]A dash (−) indicates the associated compression feature was not employed.

**Table 5. Compression Ratios for High-Resolution Initially Uncompressed Model Data**[a]

| Row | Fmt | LLC | Qnt | Rng | NSD | Size | CR | Method |
|-----|-----|-----|-----|-----|-----|------|-----|--------|
| A | N3 | - | - | $10^{37}$ | $\sim 7$ | 839.6 | 100.0 | Uncompressed |
| B | N3 | BZ1 | - | $10^{37}$ | $\sim 7$ | 581.8 | 69.3 | Bzip2 |
| C | N3 | BZ9 | - | $10^{37}$ | $\sim 7$ | 580.8 | 69.2 | Bzip2 |
| D | N7 | - | - | $10^{37}$ | $\sim 7$ | 823.2 | 98.1 | Uncompressed |
| E | N7 | DF1 | - | $10^{37}$ | $\sim 7$ | 503.7 | 60.0 | DEFLATE |
| F | N7 | DF9 | - | $10^{37}$ | $\sim 7$ | 491.3 | 58.5 | DEFLATE |
| G | N7 | - | LP | $10^{5}$ | $\sim 1$–$4$ | 413.4 | 49.2 | Linear Packing |
| H | N7 | DF1 | LP | $10^{5}$ | $\sim 1$–$4$ | 162.6 | 19.4 | Linear Packing |
| I | N7 | DF1 | BG | $10^{37}$ | $\sim 7$ | 503.6 | 60.0 | Bit Grooming |
| J | N7 | DF1 | BG | $10^{37}$ | 6 | 485.0 | 57.8 | Bit Grooming |
| K | N7 | DF1 | BG | $10^{37}$ | 5 | 427.6 | 50.9 | Bit Grooming |
| L | N7 | DF1 | BG | $10^{37}$ | 4 | 346.2 | 41.2 | Bit Grooming |
| M | N7 | DF1 | BG | $10^{37}$ | 3 | 289.6 | 34.5 | Bit Grooming |
| N | N7 | DF1 | BG | $10^{37}$ | 2 | 229.2 | 27.3 | Bit Grooming |
| O | N7 | DF1 | BG | $10^{37}$ | 1 | 161.4 | 19.2 | Bit Grooming |

[a]Notation as in Table 4.

of the values packed. The compressed packed data (Row H) reaches $\mathrm{CR} \sim 19\%$, whereas Bit Grooming ranges between $60 \geq \mathrm{CR} \geq 19\%$ for $7 \geq NSD \geq 1$ (Rows I-O).

NASA uses HDF4-format to store and distribute the third dataset tested (Table 6). Satellite-borne remote sensing datasets may be most commonly found in HDF4 format due to its early availability and the long mission duration of satellites. This dataset contains compressed (DF5) meteorological data from MERRA re-analysis (*Rienecker et al.*, 2011) on a medium resolution (latitude $\times$ longitude $= 144 \times 288$, 41472 gridpoints) grid and is 244 MB compressed (Row A) and 617–694 MB uncompressed (Rows D3–D4). *bzip2*-compression has no effect on the dataset as distributed in HDF4-format (Row B). However, converting from HDF4-format to netCDF4-format reduces its size by 13% to $\mathrm{CR} \sim 87\%$ (Rows D1–D2). Neither of these formats affords any help to *bzip2* (Rows B2–C). The uncompressed data occupies 10% less space in netCDF3- than in netCDF4-format (Rows D3–D4). The HDF5-implementation of DEFLATE yields moderately more dynamic range ($91\% \geq \mathrm{CR} \geq 85$) (Rows E–F) than in the previous two datasets. The reasons for this are unclear. Packing once again yields a 50% reduction relative to the uncompressed dataset size (Row G), and compressing that yields $\mathrm{CR} \sim 55\%$ (Row H). Bit Grooming yields $92 \geq \mathrm{CR} \geq 41\%$ for $7 \geq NSD \geq 1$ (Rows I–O).

**Table 6. Compression Ratios for High-Resolution Initially Compressed Observed HDF4 Data**

| Row | Fmt | LLC | Qnt | Rng | NSD | Size | CR | Method |
|-----|-----|-----|-----|-----|-----|------|-----|--------|
| A | H4 | DF5 | - | $10^{37}$ | ~7 | 244.3 | 100.0 | DEFLATE |
| B1 | H4 | BZ1 | - | $10^{37}$ | ~7 | 244.7 | 100.1 | Bzip2 |
| D1 | N4 | DF5 | - | $10^{37}$ | ~7 | 214.5 | 87.8 | DEFLATE |
| D2 | N7 | DF5 | - | $10^{37}$ | ~7 | 210.6 | 86.2 | DEFLATE |
| B2 | N4 | BZ1 | - | $10^{37}$ | ~7 | 215.4 | 88.2 | Bzip2 |
| C | N4 | BZ9 | - | $10^{37}$ | ~7 | 214.8 | 87.9 | Bzip2 |
| D3 | N3 | - | - | $10^{37}$ | ~7 | 617.1 | 252.6 | Uncompressed |
| D4 | N4/7 | - | - | $10^{37}$ | ~7 | 694.0 | 284.0 | Uncompressed |
| E | N4/7 | DF1 | - | $10^{37}$ | ~7 | 223.2 | 91.3 | DEFLATE |
| F | N4/7 | DF9 | - | $10^{37}$ | ~7 | 207.3 | 84.9 | DEFLATE |
| G | N4/7 | - | LP | $10^{5}$ | ~1–4 | 347.1 | 142.1 | Linear Packing |
| H | N4/7 | DF1 | LP | $10^{5}$ | ~1–4 | 133.6 | 54.7 | Linear Packing |
| I | N4/7 | DF1 | BG | $10^{37}$ | ~7 | 223.1 | 91.3 | Bit Grooming |
| J | N4/7 | DF1 | BG | $10^{37}$ | 6 | 225.1 | 92.1 | Bit Grooming |
| K | N4/7 | DF1 | BG | $10^{37}$ | 5 | 221.4 | 90.6 | Bit Grooming |
| L | N4/7 | DF1 | BG | $10^{37}$ | 4 | 201.4 | 82.4 | Bit Grooming |
| M | N4/7 | DF1 | BG | $10^{37}$ | 3 | 185.3 | 75.9 | Bit Grooming |
| N | N4/7 | DF1 | BG | $10^{37}$ | 2 | 150.0 | 61.4 | Bit Grooming |
| O | N4/7 | DF1 | BG | $10^{37}$ | 1 | 100.8 | 41.3 | Bit Grooming |

NASA uses HDF5-format to store and distribute the fourth dataset tested (Table 7) which is representative of current storage practices. HDF5 and netCDF4 are used by all new satellite missions to our knowledge. This dataset contains compressed (DF5) satellite retrievals, a swath from the OMI instrument (*Krotkov et al.*, 2008), in a curvilinear (Time × Cross-track = 1644 × 60, 98000 gridpoints) grid and is 30 MB compressed (Row A) and 50 MB uncompressed (Row D2). The dataset can be converted directly to netCDF4 (Row D1) at no additional cost in storage. However, it cannot be converted to netCDF3 because it uses so-called "enhanced" features (such as hierarchical groups) available only in netCDF4/HDF5. Once again the already-compressed data are insensitive to the level of DEFLATE (Rows E–F). Packing reduces the uncompressed size by nearly 50% (Row G), and compressing that yields CR $\sim 44\%$. Bit Grooming yields $100 \geq \text{CR} \geq 33\%$ for $7 \geq NSD \geq 1$ (Rows I–O).

**Table 7. Compression Ratios for Initially Compressed HDF5 data**

| Row | Fmt | LLC | Qnt | Rng | NSD | Size | CR | Method |
|-----|-----|-----|-----|-----|-----|------|-----|--------|
| A | H5 | DF5 | - | $10^{37}$ | ~7 | 29.5 | 100.0 | DEFLATE |
| B1 | H5 | BZ1 | - | $10^{37}$ | ~7 | 29.3 | 99.6 | Bzip2 |
| D1 | N4 | DF5 | - | $10^{37}$ | ~7 | 29.5 | 100.0 | DEFLATE |
| B2 | N4 | BZ1 | - | $10^{37}$ | ~7 | 29.3 | 99.6 | Bzip2 |
| C | N4 | BZ9 | - | $10^{37}$ | ~7 | 29.3 | 99.4 | Bzip2 |
| D2 | N4 | - | - | $10^{37}$ | ~7 | 50.7 | 172.3 | Uncompressed |
| E | N4 | DF1 | - | $10^{37}$ | ~7 | 29.8 | 101.3 | DEFLATE |
| F | N4 | DF9 | - | $10^{37}$ | ~7 | 29.4 | 99.8 | DEFLATE |
| G | N4 | - | LP | $10^{5}$ | ~1–4 | 27.7 | 94.0 | Linear Packing |
| H | N4 | DF1 | LP | $10^{5}$ | ~1–4 | 12.9 | 43.9 | Linear Packing |
| I | N4 | DF1 | BG | $10^{37}$ | ~7 | 29.7 | 100.7 | Bit Grooming |
| J | N4 | DF1 | BG | $10^{37}$ | 6 | 29.7 | 100.8 | Bit Grooming |
| K | N4 | DF1 | BG | $10^{37}$ | 5 | 27.3 | 92.8 | Bit Grooming |
| L | N4 | DF1 | BG | $10^{37}$ | 4 | 23.8 | 80.7 | Bit Grooming |
| M | N4 | DF1 | BG | $10^{37}$ | 3 | 20.3 | 69.0 | Bit Grooming |
| N | N4 | DF1 | BG | $10^{37}$ | 2 | 15.1 | 51.2 | Bit Grooming |
| O | N4 | DF1 | BG | $10^{37}$ | 1 | 9.9 | 33.6 | Bit Grooming |

## 4    Discussion

PPC algorithms preserve all significant digits of every value. The Bit Grooming (NSD) algorithm uses a theoretical approach (3.32 bits per base-10 digit), tuned and tested to ensure the *worst* case quantization error is less than half the value of the minimum increment in the least significant digit. The Decimal Rounding (DSD) algorithm uses floating-point math to round 5 each value optimally so that it has the maximum number of zeroed bits that preserve the specified precision.

While Bit Grooming works on top of any lossless compression technique, we demonstrated it with the DEFLATE algorithm (*Deutsch*, 2008) which is free and ubiquitous. Byte-stream compression techniques such as DEFLATE (which is accessible through the netCDF4/HDF5 interfaces) always compress strings of zeros and of ones more efficiently than random digits. We expect the additional compression achieved by Bit Grooming to remain roughly the same with different underlying lossless 10 compression techniques.

## 4.1 Comparison of Lossy Compression Techniques

Factors influencing the choice of lossy compression technique include precision, accuracy, dynamic range, compression ratio, and portability (*Silver and Zender*, 2016). Section 3 evaluates Bit Grooming performance alongside linear packing, a widely used, well-known lossy compression method. Packing four-byte SP floating point data into two-byte integers produces a compression ratio $CR \sim 50\%$ relative to uncompressed data (Tables 4–7, Row G). Lossless compression more than halves that CR, so that linear Packing followed by DEFLATE achieves $\sim 26\% \geq CR \geq 19\%$ (Row H) relative to uncompressed data. All other things being equal, a competitive lossy compression algorithm should produce a comparable $CR$ to be considered as a sensible option to Packing plus DEFLATE. For the tested datasets, Bit Grooming produces $43 \geq CR \geq 21\%$ for $NSD = 2$ and $29 \geq CR \geq 15\%$ for $NSD = 1$ (Rows I-O), relative to uncompressed data. Thus Bit Grooming is only competitive with compressed Packing if used aggressively (i.e., preserving only 1 or 2 digits) and/or if other factors are considered as important as CR.

These other factors may include the greater transparency, dynamic range, and guaranteed precision of Bit Grooming relative to Packing. Regarding transparency, Bit-Groomed data is valid IEEE floating point immediately suitable for analysis and plotting, whereas Packed data must first be unpacked and reconstituted into intelligible floating point data. Hence Bit-Groomed data are more portable than Packed data.

Another important consideration is precision. Bit Grooming guarantees that its lossy quantization will preserve a specified number of (decimal) significant digits. Packing into two-byte integers *always* provides 16 bits for discretization, which can potentially yield the same precision as Bit Grooming with $nsd = 4$. However, as described in Section 2.1, linear packing guarantees $nsd \gtrsim 4$ precision only for the single greatest decade of unpacked values. Unpacked values of lesser absolute magnitude lose approximately one guaranteed significant digit per decade. By contrast, Bit Grooming guarantees the specified minimum precision level over the entire IEEE range. Other types of packing, e.g., logarithmic packing or "layer packing" can alleviate though not eliminate precision issues that affect linear packing (*Silver and Zender*, 2016). However, only linear packing is a netCDF convention (*Rew et al.*, 2005). Thus other forms of packing are less portable than linear packing which (as mentioned above) is itself less portable than Bit Grooming.

In terms of range, Bit Grooming has the same dynamic ranges as IEEE SP and DP data, $\sim 10^{37}$ and $\sim 10^{308}$, respectively. Linear Packing into two-byte integers (the usual case) reduces the dynamic range to $2^{16} - 1 = 65,535$ discretely representable values that lay in a five-decade cluster within the IEEE range. The greater range of Bit Grooming relative to Packing ($\sim 10^{37}$ vs. $10^{5}$) favors it for GSMM fields that span multiple orders of magnitude, such as aerosol number concentrations, pressure, solar heating rates, and (some) tracer mixing ratios.

## 4.2 Implementation in NCO

Offering multiple quantization and compression algorithms with a consistent and simple interfaces is important so that users can easily find the algorithm that best suits their needs. This section describes the NCO implementation of the three quantization and single lossless compression algorithm that NCO exposes to user control. We focus on the new PPC algorithms

(Bit Grooming and Decimal Rounding) whose characteristics are the subject of most of this study, but we begin with a brief summary of the DEFLATE and Packing implementations that have been in NCO for 10–20 years. NCO triggers lossless DEFLATE compression with the $-\mathtt{L}$ switch followed by a compression level argument on a scale from 0 (no compression) to 9 (full compression, much slower):

`ncks -L 5 in.nc out.nc # DEFLATE lossless compression level 5`

The NCO operator `ncpdq` performs Packing quantization:

`ncpdq in.nc out.nc # Pack Data Quickly (quantization)`

NCO implements numerous packing policies (which variables should be packed) and packing maps (which datatype should a higher-precision datatype be stored as). The Users Guide (*Zender*, 2016a) contains a full description of policies and maps.

Packing followed by lossless compression is simple and yields the most impressive compression ratios in Tables 4–7.

`ncpdq -L 5 in.nc out.nc # Pack then compress`

Although Bit Grooming instantly reduces data precision, on-disk storage reduction occurs only once the data are compressed either internally (e.g., by netCDF) or externally (by a user-supplied mechanism). It is straightforward to compress data internally using the built-in compression and decompression supported by netCDF4/HDF5. For convenience, NCO automatically

activates file-wide DEFLATE deflation level one (i.e., $-\mathtt{L}$ $\mathtt{1}$) when PPC is requested for any variable in a netCDF4 output file. This makes PPC easier to use, since the user need not explicitly specify deflation. Any explicitly specified deflation (including no deflation, or $-\mathtt{L}$ $\mathtt{0}$ with NCO) overrides the PPC deflation default. If the output file is netCDF3 format, NCO emits a message that suggests internal netCDF4 or external netCDF3 compression. netCDF3 files compressed by an external utility such as gzip accrue approximately the same benefits (shrinkage) as netCDF4, although with netCDF3 the user or provider

must uncompress (e.g., gunzip) the file before accessing the data. There is no storage benefit to rounding numbers and storing them in netCDF3 files unless such custom compression/decompression is employed. Without compression, one may as well maintain the undesired precision.

NCO users can invoke PPC with the long option `--ppc` *var=prc*, or give the same arguments to the synonyms `--precision_preserving_compression`, or to `--quantize`. Here *var* is the variable to quantize, and *prc* is the

precision. NCO assumes that *prc* specifies Bit Grooming (i.e., NSD precision) so, e.g., `T=2` means $nsd = 2$. In NCO, users may prepend *prc* with a decimal point to specify decimal rounding (i.e., DSD precision), e.g., `T=.2` means $dsd = 2$. Bit Grooming precision must be specified as a positive integer. Rounding precision may be a positive or negative integer; and is specified as the negative base 10 logarithm of the desired precision, in accord with common usage. For example, specifying `T=.3` or `T=.-2` tells the Decimal Rounding algorithm to store only enough bits to preserve the value of $T$ rounded to the nearest

thousandth or hundred, respectively.

NCO users can specify the precision of an entire dataset with many variable in one simple command. Setting *var* to `default` has the special meaning of applying the associated PPC algorithm to all normal floating point variables. The exceptions, i.e., variables *not affected* by `default`, include integer and non-numeric atomic types, dimensional coordinates (such

as longitude, latitude), and, in accord with the CF Metadata Convention (*Gregory*, 2003; *Eaton et al.*, 2016), variables mentioned in the `bounds`, `climatology`, or `coordinates` attributes of any variable. These exceptions prevent the coordinate grid itself, and the variables needed to describe it, from losing precision. Usually the coordinate grid is known to much higher precision than the fields stored on the grid. NCO applies PPC to coordinate grid variables only if those variables are explicitly

specified (i.e., not with the `default=`*prc* mechanism. NCO applies PPC to integer-type variables only if those variables are explicitly specified (i.e., not with the `default=`*prc*, and only if the DSD algorithm is invoked with a negative *prc*. To prevent PPC in NCO from applying to certain non-coordinate variables (e.g., `gridcell_area` or `gaussian_weight`), explicitly specify a precision exceeding 7 (for `NC_FLOAT`) or 15 (for `NC_DOUBLE`) for those variables. Since these are the maximum representable precisions in decimal digits, NCO *turns-off* PPC (i.e., does nothing) when more precision is requested.

NCO users access PPC through a single switch, `--ppc`, repeated as many times as necessary. To request Bit Grooming only for variable *u* use, e.g.,

```
ncks -7 --ppc u=2 in.nc out.nc
```

The output file will preserve two significant digits of *u*. The options `-4` or `-7` ensure a netCDF4-format output (regardless of the input file format) to support internal compression. NCO recommends though does not require writing netCDF4 files after PPC.

However, for conciseness the `-4`/`-7` switches are omitted in subsequent examples. To maintain data-processing provenance, NCO attaches attributes that indicate the algorithm used and degree of precision retained for each variable affected by PPC. The Bit Grooming (i.e., NSD) and Decimal Rounding (i.e., DSD) algorithms store the attributes `number_of_significant_digits` and `least_significant_digit`[2], respectively. It is safe to attempt PPC on input that has already been rounded. Variables can be made rounder, not sharper, i.e., variables cannot be "un-rounded". Thus PPC attempted on an input variable with

an existing PPC attribute proceeds only if the new rounding level exceeds the old, otherwise no new rounding occurs (i.e., a "no-op"), and the original PPC attribute is retained rather than replaced with the newer value of *prc*.

To request, say, five significant digits (*nsd* = 5) for all fields, except, say, wind speeds *u* and *v* which are only known to integer values (*dsd* = 0) in the supplied units, use `--ppc` twice:

```
ncks --ppc default=5 --ppc u,v=.0 in.nc out.nc
```

To preserve five digits in all variables except coordinate variables and *u* and *v*, use the `default` option and separately specify the exceptions:

```
ncks --ppc default=5 --ppc u,v=20 in.nc out.nc
```

Specify `--ppc` option any number of times to support varying precision types and levels. Each option may aggregate all the variables with the same precision:

```
ncks --ppc p,w,z=5 --ppc q,RH=4 --ppc T,u,v=3 in.nc out.nc
```

---

[2] The nc3tonc4 tool by J. Whitaker adds the same attribute.

This type of per-variable approach to PPC may yield the best balance of precision and compression. It does require that the dataset producer understand the intrinsic precision of each variable treated in a non-default manner. For convenience, variable names may be extended regular expressions. This simplifies generating lists of related variables:

```
ncks --ppc Q.?=5 --ppc FS.?,FL.?=4 --ppc RH=.3 in.nc out.nc
```

## 5 Conclusions

We introduced a new lossy and precision-preserving compression (PPC) algorithm called Bit Grooming, and evaluated it against its nearest cousin, Bit Shaving, as well as against Packing and lossless techniques. Bit Grooming replaces the (unwanted) least significant bits of the IEEE significand with a string of identical values that alternates between zeroes and ones for consecutive elements of an array. We quantified the trade-offs involved in the choice of lossy packing technique for four climate-related datasets. We found that PPC compression reduces the volume of single-precision compressed data by roughly 10% per decimal digit quantized (or "groomed") after the third such digit. Bit Grooming reduces the storage space required by initially uncompressed and compressed climate data by 25–80% and 5–65%, respectively, for single-precision values (the most common case for climate data) quantized to retain 1–5 decimal digits of precision. Bit Groomed and Bit Shaved data are equally efficiently compressed, and Bit Grooming eliminates undesirable statistical artifacts of Bit Shaving. By alternately using zero and one as the fill-bit, Bit Grooming produces no mean absolute bias whereas Bit Shaving is negatively biased.

The lossy technique of linear Packing, followed by lossless compression, produces significantly better compression ratios than PPC algorithms like Bit Grooming for most precision levels. Bit Grooming yields comparable to or better compression than Packing only when retaining one or two significant digits of precision. Packing, however, can only encode values from a much smaller dynamic range than Bit Grooming, and its guaranteed precision degrades rapidly (one digit per decade) outside the largest decade of values to be quantized. Moreover, packed data requires additional software overhead to unpack. Bit Grooming, in contrast, works on all ranges of floating point values, has well-defined and guaranteed precision, and requires no additional software interface to read. By understanding the trade-offs between precision, statistical accuracy, numerical range and storage space of common lossy packing techniques, producers can make better decisions regarding how much precision to archive in their datasets, and how to discard the false precision.

## Code availability

NCO source code is available from GitHub at https://github.com/nco. The NCO software version 4.6.1 (*Zender*, 2016b) used to produce this paper is permanently archived with DOI 10.5281/zenodo.61341, though any version since 4.4.8 should be functionally equivalent with regards to features described here. NCO executables are available on most modern Linux and OS X systems using standard commands (apt-get install nco, brew install nco, conda install -c conda-forge nco, dnf install nco, port install nco, yum install nco). Additional binaries are available for easy installation, see the homepage http://nco.sf.net for more details. Detailed documentation and help pages are also at http://nco.sf.net. The Supplement details the commands and datasets necessary to reproduce the results.

*Acknowledgements.* Two anonymous reviewers and J. D. Silver provided helpful comments that improved the quality of this manuscript. R. Signell originally suggested we investigate Decimal Rounding. Supported by NASA ACCESS NNX12AF48A and NNX14AH55A and by DOE ACME DE-SC0012998.

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

**Supplement**

This supplement details the commands and datasets necessary to reproduce the results tabulated in the paper. For Tables 1–2, first place the exact value $\pi$ in a variable named, say, *pi* in a netCDF file named, say, in.nc. (Alternatively, use the file in.nc

that comes with NCO). Then apply Bit Grooming and Decimal rounding as follows:

```
# Define pi
ncap2 -s 'pi=3.141592653589793238462643383279502929' in.nc in.nc
# Bit Groom to every level from 1 to 9 significant digits
ncks -v pi --ppc pi=1 in.nc nsd1.nc
ncks -v pi --ppc pi=2 in.nc nsd2.nc
ncks -v pi --ppc pi=3 in.nc nsd3.nc
ncks -v pi --ppc pi=4 in.nc nsd4.nc
ncks -v pi --ppc pi=5 in.nc nsd5.nc
ncks -v pi --ppc pi=6 in.nc nsd6.nc
ncks -v pi --ppc pi=7 in.nc nsd7.nc
ncks -v pi --ppc pi=8 in.nc nsd8.nc
ncks -v pi --ppc pi=9 in.nc nsd9.nc
# Decimal rounding to 2 significant decimal places
ncks -v pi --ppc pi=.2 in.nc dsd2.nc
# Print to sixteen decimals
ncks -v pi -s %20.16e -C -H nsd1.nc
```

Many sites like http://www.h-schmidt.net/FloatConverter/IEEE754.html show the IEEE binary format of the resulting decimal numbers.

These instructions produce the statistical evaluation of Bit Grooming vs. Bit Shaving in Table 3.

```
   # Convert MERRA assimilation downloaded from NASA from HDF to netCDF
   # and extract temperature T
   ncks -3 -v T MERRA300.prod.assim.inst3_3d_asm_Cp.20130601.hdf T.nc
 # Delete extraneous packing information
   ncatted -a scale_factor,,d,, -a add_offset,,d,, T.nc
   # Copy MERRA T into SP and DP PPC input files
   # Use separate variable name for each Bit Grooming level
   # SP (Single Precision):
ncap2 -s 'ppc=T;nsd1=nsd2=nsd3=nsd4=nsd5=nsd6=nsd7=ppc' T.nc ppc_in.nc
   # DP (Double Precision):
   ncap2 -s 'ppc=double(T);nsd1=nsd2=nsd3=nsd4=nsd5=nsd6=nsd7=ppc' \
         T.nc ppc_in.nc
   # Artificial SP dataset
ncap2 -s 'defdim("dmn",1000000);ppc=float(array(1.0,1.e-6,$dmn))' \
         -s 'nsd1=nsd2=nsd3=nsd4=nsd5=nsd6=nsd7=ppc' in.nc ppc_in.nc
   # Artificial DP dataset
   ncap2 -s 'defdim("dmn",1000000);ppc=array(1.0,1.e-6,$dmn);' \
         -s 'nsd1=nsd2=nsd3=nsd4=nsd5=nsd6=nsd7=ppc' in.nc ppc_in.nc
   # Bit Groom input dataset
   ncks --ppc nsd1=1 --ppc nsd2=2 --ppc nsd3=3 --ppc nsd4=4 --ppc nsd5=5 \
         --ppc nsd6=6 --ppc nsd7=7 ppc_in.nc ppc_out.nc
   # Decimal Round input dataset
ncks --ppc nsd1=.1 --ppc nsd2=.2 --ppc nsd3=.3 --ppc nsd4=.4 \
         --ppc nsd5=.5 --ppc nsd6=.6 --ppc nsd7=.7 ppc_in.nc ppc_out.nc

   # Subtract quantized from exact data
   ncbo ppc_out.nc ppc_in.nc ppc_dff.nc
# Ratios of biases to exact data
   ncbo -y dvd ppc_dff.nc ppc_in.nc ppc_rat.nc
   # Multiply biases by scale factor for easy intercomparison
   ncap2 -s 'nsd1*=10;nsd2*=100;nsd3*=1000;nsd4*=10000;nsd5*=100000;' \
         -s 'nsd6*=1000000;nsd7*=10000000' ppc_rat.nc ppc_rat_scl.nc
# Compute statistics of biases
```

```
   ncwa -y avg ppc_rat_scl.nc ppc_avg.nc # Mean bias
   ncwa -y max ppc_rat_scl.nc ppc_max.nc # Maximum bias
   ncwa -y min ppc_rat_scl.nc ppc_min.nc # Minimum bias
   ncwa -y mabs ppc_rat_scl.nc ppc_mabs.nc # Maximum absolute bias
 ncwa -y mebs ppc_rat_scl.nc ppc_mebs.nc # Mean absolute bias
   ncwa -y mibs ppc_rat_scl.nc ppc_mibs.nc # Minimum absolute bias
```

These instructions produce the compression ratios shown in Tables 4–7. The indicated files (total size $\sim 1.2\,\mathrm{GB}$) are available from http://figshare.com after contacting the author (zender at uci dot edu). Run the indicated commands on each input file and compute the compression ratio as the output file-size divided by the initial file-size.

```
# Tables 4-7
   fl=dstmch90_clm.nc
   fl=famipc5_ne30_v0.3_00003.cam.h0.1979-01.nc
   fl=MERRA300.prod.assim.inst3_3d_asm_Cp.20130601.hdf
   fl=OMI-Aura_L2-OMIAuraSO2_2012m1222-o44888_v01-00-2014m0107t114720.h5
   # Use ls to obtain filesize for output files
   # Compute compression ratio as Row A divided by output filesize
   ls -l ${fl} # Row A
   bzip2 -1 -f ${fl} # Row B
bzip2 -9 -f ${fl} # Row C
   ncks -7 -L 0 ${fl} foo.nc # Row D
   ncks -7 -L 1 ${fl} foo.nc # Row E
   ncks -7 -L 9 ${fl} foo.nc # Row F
   ncpdq -7 -L 0 ${fl} foo.nc # Row G
ncpdq -7 -L 1 ${fl} foo.nc # Row H
   ncks -7 -L 1 --ppc default=7 ${fl} foo.nc # Row I
   ncks -7 -L 1 --ppc default=6 ${fl} foo.nc # Row J
   ncks -7 -L 1 --ppc default=5 ${fl} foo.nc # Row K
   ncks -7 -L 1 --ppc default=4 ${fl} foo.nc # Row L
ncks -7 -L 1 --ppc default=3 ${fl} foo.nc # Row M
   ncks -7 -L 1 --ppc default=2 ${fl} foo.nc # Row N
   ncks -7 -L 1 --ppc default=1 ${fl} foo.nc # Row O
```