# Peer review of "Bit Grooming: Statistically accurate precision-preserving quantization with compression, evaluated in the netCDF Operators (NCO, v4.4.8+)"

_Geoscientific Model Development, 2016_

## Referee Comment (RC1) · Anonymous Referee #1 · 19 May 2016

This paper is a short clear paper describing a simple but useful concept, that of "Bit Grooming", a technique that allows more compact storage of scientific data, preserves an unbiased mean, and allows the data creator to store just as much precision as is justified. The paper describes some common data compression techniques, and discusses the pros and cons of various techniques. Although many of these concepts have previously appeared in the literature, these techniques are still not widely known in the field, and this paper provides a nice review of the state of the art, which should prove useful to the community. In addition, it introduces a simple but novel concept of "Bit Grooming".

A few minor comment/questions only:

In the Abstract on line 13, it is mentioned that Bit Grooming produces storage reductions comparable to other quantization techniques such as linear packing when "used aggressively". Is this always true?

On line 22, the statement that begins "False precision can mislead..." and the following sentences express a concept that should be captured in the abstract. This is the real strength of this approach: turning useless precision into something that is (a) more honest, and (b) saves space!

The "eight-hundred pound gorilla" example is cute, but perhaps a better example would be something less cute and ordinary, such as a "liter of milk" or something.

It's great that the source code is provided on Github. Kudos to the authors for making the code truly open source!

---

## Referee Comment (RC2) · Anonymous Referee #2 · 11 Jul 2016

The core contribution of this paper appears to be the level of compression achieved while retaining a high degree of dynamic range, as well as statistical properties of resulting data.

This contribution, compared to other methods, is only clearly articulated in the sentence spanning pages 9-11.

Tables 4-7, with some interpretation, are good at conveying the relative resultant size after applying the algorithms examined. This may be a good place to bring together, and highlight, the interplay between the data size and precision achieved at that size.

[Figure]

The number of significant digits is already presented for the Bit Groomer methods. Could this be added for the other methods, either in theory or on a particular data set? Additionally can the dynamic range, or number of bits remaining in the mantissa?

————————————————

---

## Author Comment (AC2) · 30 Jul 2016

(Reviewer's comments/questions are in *italics* and my responses are interspersed in plain text.)

I thank the Reviewer for their comments. The Reviewer's questions indicate that the submitted manuscript did not give enough background on the precision and range of the methods employed besides Bit Grooming, and did not adequately intercompare the trade-offs of Bit Grooming with the trade-offs of the other methods, linear Packing in particular. The revised manuscript, which is attached as a PDF to this response,

includes a fuller explanation of the range and precision of linear Packing. This made it easier to describe the trade-offs between compression ratio and precision incurred by Bit Grooming in comparison to Packing.

*The core contribution of this paper appears to be the level of compression achieved while retaining a high degree of dynamic range, as well as statistical properties of resulting data. This contribution, compared to other methods, is only clearly articulated in the sentence spanning pages 9–11.*

Agreed. The revised manuscript now presents more clearly the trade-offs between size and precision for linear Packing in Section 2.2, and the trade-offs for Bit Grooming in Section 2.3. While these trade-offs are still noted in the results in Section 3.3, the inter-comparison of these trade-offs (which the Referee suggests is the core contributions of the paper) is now concentrated in the the newly created Section 4.1 "Comparison of Lossy Compression Techniques".

*Tables 4–7, with some interpretation, are good at conveying the relative resultant size after applying the algorithms examined. This may be a good place to bring together, and highlight, the interplay between the data size and precision achieved at that size.*

Agreed. As described above, we now bring together the discussion of interplay between size, range, and compression in the newly created Section 4.1. In addition, Tables 4–7 are now easier to interpret. Separate columns clarify the lossless compression method (column LLC), the quantization method (column Qnt), and the overall compression method (column Method).

*The number of significant digits is already presented for the Bit Groomer methods. Could this be added for the other methods, either in theory or on a particular data set?*

The NSD column in Tables 4–7 now includes the precision for all methods.

*Additionally can the dynamic range, or number of bits remaining in the mantissa?*

Precision and dynamic range are, for floating point values, determined by bits in the

mantissa and exponent, respectively. This is the case for Bit Grooming. Since packed data are integers and have no exponent, their integer bits determine both their unpacked precision and their dynamic range. Hence we interpret the Referee's questions as asking whether both precision and dynamic range can be added to the Tables. The short answer is Yes, and the revised manuscript includes this information in Tables 4–7.

The new Range column specifies the dynamic range for each method. And the NSD column has been completely filled-in to show the number of significant digits for all methods, not just Bit Grooming. The number of bits retained (in contrast to digits) is described in Section 2.2 and 2.3 for Packing and Bit Grooming, respectively.

**Errata:**

1. During editing I identified and corrected a numerical mistake. The original manuscript erroneously multiplied the exponents rather than the mantissas by two to estimate the dynamic range of IEEE SP and DP from their maximal values. The correction in the revised manuscript changes

   "The dynamic ranges of SP and DP numbers are $\sim 10^{74}$ and $\sim 10^{616}$, respectively, whereas data packed linearly into two-byte and four-byte integers have dynamic ranges of $\sim 10^5$ and $\sim 10^{10}$, respectively.",

   to

   "The dynamic ranges of SP and DP numbers are $\sim 10^{37}$ and $\sim 10^{308}$, respectively, whereas data packed linearly into two-byte and four-byte integers have dynamic ranges of $\sim 10^5$ and $\sim 10^{10}$, respectively.".

   Changing the exponents by a factor of two does not qualitatively alter the results or conclusions of the manuscript.

2. The revised manuscript now contains the citable references to the four datasets that are intercompared, and the datasets have been made available online (figshare.com) as described in the Supplement.

3. The revised manuscript contains many minor wording changes that characterize the precision and dynamic range of linear Packing, as requested by the Referee, and how these characteristics compare to those for Bit Grooming.

Please also note the supplement to this comment:
http://www.geosci-model-dev-discuss.net/gmd-2016-63/gmd-2016-63-AC2-supplement.pdf
* * *
[Figure]

**Supplement:**

[revised manuscript text omitted]

```

---

## Author Response (AR1)

**Reviews & Responses**

Referee #1 Comments Received 20160518:

(Reviewer's comments/questions are in *italics* and my responses are interspersed in plain text.)

1. *This paper is a short clear paper describing a simple but useful concept, that of "Bit Grooming", a technique that allows more compact storage of scientific data, preserves an unbiased mean, and allows the data creator to store just as much precision as is justified. The paper describes some common data compression techniques, and discusses the pros and cons of various techniques. Although many of these concepts have previously appeared in the literature, these techniques are still not widely known in the field, and this paper provides a nice review of the state of the art, which should prove useful to the community. In addition, it introduces a simple but novel concept of "Bit Grooming". A few minor comment/questions only:*

   I thank the Reviewer for their thoughtful comments. I share the Reviewer's perspective that these techniques are underappreciated in the Geosciences, and am glad to help rectify that in a small way.

2. *In the Abstract on line 13, it is mentioned that Bit Grooming produces storage reductions comparable to other quantization techniques such as linear packing when "used aggressively". Is this always true?*

   The wording of this question makes it important to clarify for others that the manuscript asserts that it is Bit Grooming (not, e.g., Packing) that must be used aggressively to match compression ratios (CRs) produced by other techniques (e.g., Packing). Standard Packing (float32→short16) of float-dominated data always produces CR of about 50%. This CR is intrinsic to the Packing algorithm and applies to any float-dominated dataset (the only type of dataset this manuscript discusses).

   Our results show lossless compression further reduces Packing-assisted CRs to about 20% (relative to uncompressed data). In every case tested Bit Grooming must be used aggressively (i.e., preserve at most 2 significant digits) to match or best these CRs. In one case (Table 4) Packing produces better CRs than Bit-Grooming at all precision levels, and in the other cases (Tables 5–7) Bit-Grooming with $NSD = 1$ beats Packing. The differences in CRs produced by Packing and by aggressive Bit-Grooming are generally within 10%, which I consider to be "comparable" performance. I chose the test data to be representative, and know of no real-world float-dominated datasets where Bit Grooming CRs could match or best Packing CRs unless Bit Grooming were used this aggressively. So the answer

to your original question, as I interpret it, is "Yes". The manuscript now clarifies this in the abstract:

"When used aggressively (i.e., preserving only 1–2 decimal digits of precision), Bit Grooming produces storage reductions comparable to other quantization techniques such as linear packing."

and in Section 4.1:

"Thus Bit Grooming is only competitive with compressed Packing if used aggressively (i.e., preserving only 1 or 2 digits) and/or if other factors are considered as important as CR. These other factors may include the greater transparency, dynamic range, and guaranteed precision of Bit Grooming relative to Packing."

3. *On line 22, the statement that begins "False precision can mislead..." and the following sentences express a concept that should be captured in the abstract. This is the real strength of this approach: turning useless precision into something that is (a) more honest, and (b) saves space!*

Agreed. The revised manuscript abstract now includes roughly the same content as the three sentences you refer to. As a result the revised abstract now has a longer first paragraph (second paragraph is unchanged):

"Geoscientific models and measurements generate false precision (scientifically meaningless data bits) that wastes storage space. False precision can mislead (by implying noise is signal) and be scientifically pointless, especially for measurements. By contrast, lossy compression can be both economical (save space) and heuristic (clarify data limitations) without compromising the scientific integrity of data. Data quantization can thus be appropriate regardless of whether space limitations are a concern. We introduce, implement, and characterize a new lossy compression scheme suitable for IEEE floating-point data. Our new Bit Grooming algorithm alternately shaves (to zero) and sets (to one) the least significant bits of consecutive values to preserve a desired precision. This is a symmetric, two-sided variant of an algorithm sometimes called Bit Shaving which quantizes values solely by zeroing bits. Our variation eliminates the artificial low-bias produced by always zeroing bits, and makes Bit Grooming more suitable for arrays and multi-dimensional fields whose mean statistics are important."

4. *The "eight-hundred pound gorilla" example is cute, but perhaps a better example would be something less cute and ordinary, such as a "liter of milk" or something.*

It is important that the example have more than one digit, and also that some digits be insignificant, i.e., that the quantity be recognized as an approximation that is not exact. And finally the example

must be dimensional and denominated in a standard unit like mass, volume, or time. A "liter of milk" won't work, neither will 10 or 100 liters because milk bottles are measured in exact units with high precision. I don't see the drawback of the gorilla example, which has the necessary properties. Ordinary examples can be good, and cute examples can increase readers' interest and retention.

5. *It's great that the source code is provided on Github. Kudos to the authors for making the code truly open source!*

   Thank you for appreciating the importance of this!

Referee #2 Comments Received 20160711:

(Reviewer's comments/questions are in *italics* and my responses are interspersed in plain text.)

I thank the Reviewer for their comments. The Reviewer's questions indicate that the submitted manuscript did not give enough background on the precision and range of the methods employed besides Bit Grooming, and did not adequately intercompare the trade-offs of Bit Grooming with the trade-offs of the other methods, linear Packing in particular. As far as I know, the precision and range characteristics of Packing have not been described in the published literature. Doing so required adding a few paragraphs and altering others. This made it easier to describe the trade-offs between compression ratio and precision incurred by Bit Grooming in comparison to Packing. These changes are quoted at length below and in the attached latexdiff.

1. *The core contribution of this paper appears to be the level of compression achieved while retaining a high degree of dynamic range, as well as statistical properties of resulting data. This contribution, compared to other methods, is only clearly articulated in the sentence spanning pages 9–11.*

   Agreed. The revised manuscript now presents more clearly the trade-offs between size, range, and precision for linear Packing in Section 2.1 "Packing", and the trade-offs for Bit Grooming in Section 2.2 "Bit-Grooming", the altered/added paragraphs of which are now:

[revised manuscript text omitted]

2. *Tables 4–7, with some interpretation, are good at conveying the relative resultant size after applying the algorithms examined. This may be a good place to bring together, and highlight, the interplay between the data size and precision achieved at that size.*

Agreed. As described above, we now bring together the discussion of interplay between size, range, and compression in the newly created Section 4.1. In addition, Tables 4–7 are now easier to interpret. Separate columns clarify the lossless compression method (column LLC), the quantization method (column Qnt), and the overall compression method (column Method).

3. *The number of significant digits is already presented for the Bit Groomer methods. Could this be added for the other methods, either in theory or on a particular data set?*

The NSD column in Tables 4–7 now includes the precision for all methods, as requested. Further answered in next response.

4. *Additionally can the dynamic range, or number of bits remaining in the mantissa?*

Precision and dynamic range are, for floating point values, determined by bits in the mantissa and exponent, respectively. This is the case for Bit Grooming. Since packed data are integers and have no exponent, their integer bits determine both their unpacked precision and their dynamic range. Hence we interpret the Referee's questions as asking whether both precision and dynamic range can be

added to the Tables. The short answer is Yes, and the revised manuscript includes this information in Tables 4–7. The longer answer that Packing precision depends on the distribution of values, and that the *add_offset* offset can improve the precision of packing for some but not all distributions of data. Thus Tables 4–7 report the range of best precision that Packing can guarantee for unpacked data in the general case, not for special distributions of data.

The new Range column specifies the dynamic range for each method. And the NSD column has been completely filled-in to show the number of significant digits for all methods, not just Bit Grooming. The number of bits retained (in contrast to digits) is described in Section 2.1 for Packing:

"The type conversion frees-up the IEEE754 exponent bits (8 bits for SP, and 11 bits for DP) which then contribute to the dynamic range of the packed integers (16 and 32 bits for NC_SHORT and NC_INT, respectively)."

and the number of bits for Bit-Grooming is now described more precisely in Section 2.2:

"The exact numbers of explicit mantissa bits *Nbit* retained for single and double precision values are ceil$(3.32 \times nsd) + 1$ and ceil$(3.32 \times nsd) + 2$, respectively. (The IEEE format includes a single mantissa bit that is implicit and that is not included in these counts because it consumes no memory). This is more than predicted by the simple rule that the required number of bits is $nsd \times \ln(10)/\ln(2)$. The extra bits are the (experimentally determined) overhead required to guarantee that terminal significant digits are accurate within half the minimal value of their decimal position. Once the number of bits required exceeds the IEEE SP and DP storage standards of 23 and 53 explicit mantissa bits, respectively, bitmasking is completely ineffective. This occurs at *nsd* = 6.3 and 15.4, respectively. To guarantee preserving 1–7 digits of precision, Bit Grooming must retain $5, 8, 11, 15, 18, 21$, and $25$ explicit mantissa bits, respectively. Thus Bit Grooming (and IEEE) require DP format to guarantee $nsd \geq 7$."

**Errata:**

1. The revised manuscript contains many minor wording changes that characterize the precision and dynamic range of linear Packing, as requested by Referee #2, and how these characteristics compare to those for Bit Grooming.

2. The original manuscript erroneously multiplied the exponents rather than the mantissas by two to estimate the dynamic range of IEEE SP and DP from their maximal values. The correction in the revised manuscript changes

   "The dynamic ranges of SP and DP numbers are $\sim 10^{74}$ and $\sim 10^{616}$, respectively, whereas data packed linearly into two-byte and four-byte integers have dynamic ranges of $\sim 10^5$ and $\sim 10^{10}$, respectively.",

   to

   "The dynamic ranges of SP and DP numbers are $\sim 10^{37}$ and $\sim 10^{308}$, respectively, whereas data packed linearly into two-byte and four-byte integers have dynamic ranges of $\sim 10^5$ and $\sim 10^{10}$, respectively.".

   Changing the exponents by a factor of two does not qualitatively alter the results or conclusions of the manuscript.

3. The revised manuscript now contains citations to the four datasets that are intercompared, and the datasets are now available online (http://figshare.com) as described in the Supplement. The revised manuscript also cites a new manuscript submitted to GMD by Silver and Zender that re-uses three of these datasets in benchmarking a new packing algorithm optimized to preserve higher precision.

[revised manuscript text omitted]

```

---

## Author Response (AR2)

**Response to Editor Review**

1. *Thank you for the revised submission. I concur that you have addressed the relatively minor concerns raised by the reviewers. In essence the paper is ready for publication. However, the generally excellent provenance information provided in the paper is just slightly let down by the lack of an explicit statement about the exact revision of NCO used to conduct the experiments, and a permanent archive of that version. This could easily be achieved using the GitHub Zenodo integration, or you could upload a tarball to Figshare if you prefer. Once that is done, this manuscript is ready to go.*

   I had not heard of Zenodo before yet it proved relatively easy to implement. The revised manuscript now states the DOI provided by Zenodo in the Code Availability section, and cites this release:

   "The NCO software version 4.6.1 (Zender, 2016b) used to produce this paper is permanently archived with DOI 10.5281/zenodo.61341, though any version since 4.4.8 should be functionally equivalent with regards to features described here."

2. In the interim I corrected two misleading statements in the abstract:

   "Bit Grooming reduces the storage space required by uncompressed and compressed climate data by up to 50% and 20%, respectively, for single-precision data (the most common case for climate data) ... Bit Grooming reduces the volume of single-precision compressed data by roughly 10% per decimal digit quantized (or "groomed") after the third such digit, up to a maximum reduction of about 50%."

   In fact the upper bounds (as shown in the tables for preserving 1 decimal digit of precision), are 80% and 65% (not 50% and 20%), respectively. The misleading statements were originally penned with an assumed (though unstated) choice of retaining 3–4 digits of precision. At the time I thought this was a reasonable and conservative choice but in hindsight I see this as a subjective choice that conveys incomplete information and obscures the potential power of Bit Grooming. It is more clear to state the full range of compression that can be achieved by the full range of preserved precisions. I replaced those statements in the abstract with:

[revised manuscript text omitted]

```